# TiO_2_ Passivation Layer on ZnO Hollow Microspheres for Quantum Dots Sensitized Solar Cells with Improved Light Harvesting and Electron Collection

**DOI:** 10.3390/nano10040631

**Published:** 2020-03-28

**Authors:** Zhen Li, Libo Yu, Hao Wang, Huiwen Yang, Huan Ma

**Affiliations:** 1College of Chemistry and Chemical Engineering, Hexi University, Zhangye City 734000, China; wanghao@hxu.edu.cn (H.W.); Yanghuiwen@hxu.edu.cn (H.Y.); Mahuan@tju.edu.cn (H.M.); 2Key Laboratory of Hexi Corridor Resources Utilization of Gansu, Hexi University, Zhangye City 734000, China

**Keywords:** TiO_2_ passivation layer, ZnO hollow microspheres, quantum dots sensitized solar cells

## Abstract

Light harvesting and electron recombination are essential factors that influence photovoltaic performance of quantum dots sensitized solar cells (QDSSCs). ZnO hollow microspheres (HMS) as architectures in QDSSCs are beneficial in improving light scattering, facilitating the enhancement of light harvesting efficiency. However, this advantage is greatly weakened by defects located at the surface of ZnO HMS. Therefore, we prepared a composite hollow microsphere structure consisting of ZnO HMS coated by TiO_2_ layer that is obtained by immersing ZnO HMS architectures in TiCl_4_ aqueous solution. This TiO_2_-passivated ZnO HMS architecture is designed to yield good light harvesting, reduced charge recombination, and longer electron lifetime. As a result, the power conversion efficiency (PCE) of QDSSC reaches to 3.16% with an optimal thickness of TiO_2_ passivation layer, which is much higher when compared to 1.54% for QDSSC based on bare ZnO HMS.

## 1. Introduction

The collapse of current climate because of global warming and huge demand of energy have forced us to conduct extensive research on solar cells over the past decades. Tremendous progress has been achieved in solar cell technology because of the increasing research focusing on the development of nanotechnology and new materials [1,2,3,4]. During the research development of solar cells, quantum dots sensitized solar cells (QDSSCs) have received wide attention because of their several merits including low temperature fabrication, solution-based processing, and high stability in different surroundings [5,6,7]. QDSSCs comprise three main parts: photoanode, electrolyte, and counterelectrode [8]. The photoanode usually consists of fluorine-doped tin oxide (FTO) glass, mesoporous metal oxide semiconductor (MOS) film, and quantum dots (QDs) [9,10]. Among these components of photoanode, the MOS as supporting architecture plays an important role in light utilization and charge transfer process in QDSSCs. Hence, design and decoration of an appropriate MOS architecture has been considered as an effective approach to improve light harvesting and electron transport in QDSSCs [11], enhancing the photovoltaic performance.

Various MOS materials such as ZnO, TiO_2_, and SnO_2_ are suitable for application in QDSSCs [12,13,14]. Among them, ZnO is a potential candidate material for the photoanode because of its high electron mobility (200–1000 cm^−2^ V^−1^ S^−1^) [15] and various fabrication strategies such as atomic layer deposition (ALD) and hydrothermal method, for variety of structures including 1D, 2D, and 3D [16,17,18]. The most common ZnO architecture in QDSSCs is 1D nanostructures such as nanowire, nanotube, and nanorod arrays [12,19,20], because they can provide direct pathway for electron transport to reduce the probability of charge recombination [21,22]. However, weak light scattering of 1D ZnO structure limits the further improvement of photovoltaic performance of QDSSCs. According to the Mie theory and Anderson localization of light [23], resonant scattering of light is expected for spherical particles only when the particle size is comparable to the wavelength of incident light. To tackle this challenge, MOS hollow microspheres (HMS) as photoanode architecture seem to be a good choice for strong light scattering because of size controllability and multiple light reflection inside of HMS [24]. Our group’s previous research found that ~500 nm ZnO HMS can generate strong light scattering, improving the power conversion efficiency (PCE) of Zn_x_Cd_1−x_Se QDSSC [25].

Another factor that limits ZnO application as architecture in QDSSCs is that there are multiple surface defects in ZnO [26], which increase the charge recombination and result in the drop of short-circuit current (*J*_sc_). Covering passivation layer on surface of photoanode is an effective way to lower the surface defects and suppress charge recombination [27,28]. For example, Haifeng Zhao et al. [29] modified ZnO nanorods with TiO_2_ nanoparticles, and revealed that TiO_2_ passivation layer can facilitate the deposition of CdS/CdSe QDs on TiO_2_/ZnO surfaces, and effectively suppress the charge recombination. Lou et al. [30] reported a PCE of 1.97% for QDSSCs based on ZnO nanorods passivated with TiO_2_ as a barrier layer. In view of these backgrounds, it can be found that ZnO surface modification with TiO_2_ has attracted interest of researchers to enhance photovoltaic performance of QDSSCs, because of the combination of excellent electron mobility of ZnO and high chemical stability of TiO_2_ [29].

In this investigation, we synthesized ZnO HMS as architectures for QDSSCs through carbonaceous template method, and further passivated surface of ZnO HMS with TiO_2_ by a simple immersion route in TiCl_4_ aqueous solution. The photovoltaic performance of CdSe/CdS QDSSCs has been improved by adjusting the thickness of TiO_2_ passivation layer through optimization of immersing time in TiCl_4_ aqueous solution. The coating of TiO_2_ passivation layer not only enlarges the deposition of QDs on TiO_2_-passivated ZnO HMS surfaces, but also reduces the charge recombination and prolong the electron lifetime, leading to the enhancement of light harvesting and charge collection efficiency. As a result, the PCE of QDSSCs increased from 1.54% for bare ZnO HMS to 3.16% for TiO_2_-passivated ZnO HMS with an optimal thickness of passivation layer.

## 2. Materials and Methods 

### 2.1. Materials

All the chemicals used in this work including sucrose, zinc nitrate, cadmium nitrate, selenium powder, sodium borohydride, ethylcellulose, terpinol, ethanol, titanium tetrachloride, copper sulphate, thiourea, sodium sulfide, and sulfur powder were obtained from Aladdin Co., Ltd. (Shanghai, China), and were analytical grade reagents without further purification. Other materials such as fluorine-doped tin oxide (FTO) conductive glass were purchased from Opvtech Co., Ltd. (Dalian, China). 

### 2.2. Preparation Carbonaceous Spheres Templates

The carbonaceous spheres template was obtained by a classical hydrothermal route [31]. Generally, 1.5 M sucrose aqueous solution was transferred in to Teflon autoclave filling 80% volume, then sealed the autoclave and heated at 180 °C for 8 h. After cooling down to room temperature, the autoclave was opened and the black powder was collected. The black powder was washed three times with deionized water and dried in electric oven at 80 °C for 12 h, finally getting the carbonaceous spheres templates.

### 2.3. Synthesis of ZnO Hollow Microspheres (HMS) and Fabrication of ZnO HMS Photoanode

Total of 1.0 g newly prepared carbonaceous microspheres were soaked in 20 mL 1.0 M zinc nitrate aqueous solution, after 30 min ultrasonication, the mixture was aged for 12 h at room temperature. Then the mixture was filtered, washed three times with deionized water, and dried at 80 °C for 12 h. Subsequently, the resultant product was heated to 500 °C in Muffle furnace with rate a of 5 °C/min, and was held at this temperature for 4 h, producing white ZnO hollow microspheres (HMS) powders.

The ZnO HMS photoanode was fabricated according to the method proposed in our previous report [25]. Typically, ZnO HMS (3.0 g), ethylcellulose (0.5 g), terpinol (10 mL), and ethanol (3 mL), were mixed together under magnetic stirring for 1 h, forming a viscous paste. Then the paste was doctor-bladed onto conductive surface of FTO glass (2.0 × 1.5 cm). The thickness of the ZnO HMS film was controlled to be ~15 μm, and the active area of ZnO HMS film was tuned to be 0.25 cm^2^ with aid of same size and thickness spacer. Subsequently, the film was dried in room temperature, and was heated to 500 °C for 1 h to remove any organic residuals.

### 2.4. Fabrication of TiO_2_-Passivated ZnO Hollow Microspheres Photoanode

The TiO_2_-passivated ZnO hollow microspheres photoanodes which are designated as TiO_2_@ZnO HMS were fabricated via one-step approach. First, the prepared ZnO HMS photoanode was immersed in 1.5 M TiCl_4_ aqueous solution for a period at room temperature to guarantee the Ti^4+^ precursors can be deposited on the surface of ZnO HMS. After rinsing in deionized water, TiO_2_@ZnO HMS photoanodes were obtained by a further calcining process at 450 °C for 1 h. In order to investigate the thickness effect of TiO_2_ passivation layer, the immersing time in TiCl_4_ aqueous solution was adjusted to 5 min, 10 min, and 15 min (designated 5-min, 10-min, and 15-min). 

### 2.5. Assemble of ZnS/CdSe/CdS Quantum Dots Sensitized Solar Cells

For in situ assembly of CdSe/CdS QDs, the classic successive ionic layer adsorption reaction (SILAR) method was employed. Initially, six SILAR cycles of CdS QDs were deposited on TiO_2_@ZnO HMS photoanode by dipping photoanode in 0.1 M cadmium nitrate solution and 0.1 M sodium sulfide solution in sequence according to our previous work [32]. Then seven SILAR cycles of CdSe QDs were sensitized by dipping CdS/TiO_2_@ZnO HMS photoanodes in 0.1 M cadmium nitrate solution and 0.1 NaSeH_4_ solution in sequence to form CdSe/CdS QDs-sensitized systems. Finally, a ZnS passivation layer was formed with two SILAR cycles by dipping alternatively into 0.1 M zinc nitrate and 0.1 M sodium sulfide solutions for 1 min/dip. The identical QDs systems were applied to TiO_2_@ZnO HMS (5-min, 10-min, and 15-min) and bare ZnO HMS photoanodes for comparative investigation. 

Cu_2_S/FTO was used as the counter electrode for QDSSCs, which was prepared according to previous literature [33]. Polysulfide electrolyte was prepared by 1 M sulfur and 1 M sodium sulfide in water/methanol (1:1 in volume) solution before each test. The QDSSC was assembled by sandwiching the QDs sensitized photoanode and counter electrode in an open way, and the active area of the QDSSCs was 0.25 cm^2^. Moreover, each solar cell have been repeatedly tested three times for the purpose of reproduction.

### 2.6. Characterization

The morphologies and microstructures of the products were characterized by Tecnai G2 F20 transmission electron microscope (TEM), and Quanta 450 FEG scanning electron microscopy (SEM) equipped with an energy dispersive X-ray spectrometer (EDS) for elemental scanning. The crystalline nature and structure of TiO_2_-passivated ZnO HMS was analyzed by X-ray diffraction (XRD, D/MAX—2400, Rigaku, Tokyo, Japan) using a Cu Kα source operated at 40 kV and 30 mA with scanning rate of 2°/min. The optical absorption properties of the photoanodes were recorded by a U-3900H UV-vis spectrophotometer which is equipped with integrating sphere attachment for diffuse reflection measurement. The *I*-*V* performances of the QDSSCs were obtained under illumination using a solar simulator to simulate sunlight with intensity of 100 mW cm^−2^. The incident photon to charge carrier generation efficiency (IPCE) was measured as a function of wavelength by a 150 W Xe lamp coupled with a computer-controlled monochromator. In order to reveal the influence of TiO_2_ passivation layer on charges transfer dynamics, the electrochemical impedance spectroscopy (EIS) tests of the QDSSCs were performed by CHI852C electrochemical workstation under dark at an applied bias of −0.5 V with the frequency range of 10^−1^ to 10^−5^ Hz.

## 3. Results and Discussion

The formation of carbonaceous spheres templates is the first key step for the synthesis of ZnO HMS. Figure 1a shows the SEM image of prepared carbonaceous spheres, presenting smooth surface and uniform sizes of around 800–900 nm. The spherical structure of template are also confirmed by TEM in Figure 1b, showing that the carbonaceous spheres prepared by hydrothermal process using sucrose aqueous solution are solid spheres, and the size distribution is almost in agreement with the SEM results. Based on carbonaceous spheres, the Zn^2+^ were adsorbed on their surface just by soaking them in Zn^2+^ aqueous solution. Figure 1c shows the carbonaceous spheres after soaking treatment in Zn^2+^ aqueous solution. Obviously, this treatment make the carbonaceous spheres thicker and the size of spheres becomes a little larger than bare carbonaceous spheres shown in Figure 1b, indicating the successful adsorption of Zn^2+^ on carbonaceous spheres templates. A further annealing process at 500 °C with heating rate of 5 °C/min leads to the formation of ZnO HMS. Figure 1d shows the SEM image of ZnO products; spherical structure that is composed of large amount of nanoparticles can be clearly identified. The inset of Figure 1d is the magnified SEM image of one sphere, indicating a hollow inside of ZnO spheres. The ZnO hollow microspheres (HMS) could be identified by TEM images as shown in Figure 1a and c. A number of ZnO HMS can be seen in low magnification TEM image, indicating that massive production of ZnO HMS is achievable by this template method. The TEM of ZnO HMS in high magnification offers typical hollow spherical structure with a single shell. From the SEM and TEM of ZnO HMS, it can be seen that the size distribution of ZnO HMS is around 450–600 nm, which are smaller than carbonaceous spheres templates. When heated, the template gradually changed to CO_2_, becoming smaller and smaller. Meanwhile Zn^2+^ precursors on the surface of carbonaceous spheres were oxidized into ZnO nanoparticles and compact together as the shrinkage of template, forming ZnO HMS in smaller than primitive carbonaceous spheres templates.

Figure 2 presents the TEM images of ZnO HMS passivated by TiO_2_. As displayed in Figure 2a–c, there is almost no change in hollow microsphere structure after the TiO_2_ passivation but an extra layer on surface of ZnO HMS. A bright layer with different thickness coating on shell of ZnO HMS is apparently observed in each of ZnO HMS, as shown in the insets of Figure 2a–c, indicating that TiO_2_ passivation layer are formed on surface of ZnO HMS. Figure 2a shows the thickness of passivation layer is ~10 nm for the 5-min TiO_2_@ZnO HMS. When immersing time of ZnO HMS photoanode in TiCl_4_ aqueous solution prolongs to 10-min, the thickness of TiO_2_ passivation layer increases to ~20 nm, as shown in inset of Figure 2b. A ~32 nm thickness of TiO_2_ passivation layer is obtained by further prolonging the immersing time to 15 min, as presented in the inset of Figure 2c. These TEM images of TiO_2_@ZnO HMS indicate that the thickness of passivation layer is controllable by changing the soaking duration of ZnO HMS in TiCl_4_ aqueous solution. Figure 2d presents the HRTEM of TiO_2_@ZnO HMS (10-min), showing a lattice fringe of 0.26 nm which can be ascribed to the (002) plane of hexagonal ZnO (PDF # 36-1451). It can be identified that the outmost layer of crystallites has a lattice fringe of 0.24 nm close to the (002) lattice plane of ZnO, which is recognized as the (103) plane of anatase TiO_2_ (PDF # 21-1272). Appendix A provides the XRD of TiO2-passivated ZnO HMS, apart from diffraction peaks that can be ascribed to the hexagonal ZnO, the diffraction peaks around 2θ = 24.6° and 53.1° belongs to the anatase phase of TiO2 (See Appendix A), confirming the anatase TiO2 layer formation on ZnO HMS. The TiO2 has a higher electrical conductivity, which is favorable for electron transport [29].

The surface elemental composition of the TiO_2_@ZnO HMS photoanode (10-min) obtained by immersing ZnO HMS photoanode in TiCl_4_ aqueous solution for 10 min was characterized by EDS elemental mapping, which is shown in Figure 3. Ti, O, and Zn elements are detected from the selected zone of the photoanode. These elements are uniformly distributed on the surface of photoanode, further proving the formation of TiO_2_ passivation layer on ZnO HMS photoanode. Figure 4 illustrates the formation mechanism of TiO_2_@ZnO HMS. First, ZnO HMS are doctor bladed onto the surface of FTO glass, forming ZnO HMS photoanode, then Ti^4+^ precursors are absorbed by ZnO HMS photoanode when immersed in TiCl_4_ aqueous solution. Finally, Ti^4+^ precursors are oxidized into TiO_2_ nanoparticles during the annealing process, assembling into TiO_2_ passivation layer on surface of ZnO HMS photoanode. The thickness of passivation layer are controllable by adjusting immersing duration in TiCl_4_ aqueous solution. 

Based on TiO_2_@ZnO HMS photoanode, CdSe/CdS QDSSCs were fabricated to investigate the influence of passivation layer and layer thickness on photovoltaic performances. In Appendix A presents the three times tested *J*-*V* curves CdSe/CdS QDSSCs based on ZnO HMS passivated with TiO_2_ by immersing in TiCl_4_ aqueous solution for 0 min (blank), 5 min (5-min), 10 min (10-min), and 15 min (15-min). Appendix A summarize the photovoltaic performance of solar cells, including short-circuit current density (*J*_sc_), open voltage (*V*_oc_), fill factor (*FF*), power conversion efficiency (*PCE*), and their corresponding average value and standard deviation (See Appendix A). Furthermore, Figure 5 presents the champion *J*-*V* curves of QDSSCs, and Table 1 shows the average photovoltaic performance and standard deviation extracted from *J*-*V* curves. For the QDSSC based on bare ZnO HMS (blank), the *V*_oc_, *J*_sc_, and *FF* are 0.40 V, 8.79 mA cm^−2^, and 0.44, only producing a *PCE* of 1.54%. The increment of *PCE* occurred when TiO_2_ passivation layer formed on surface of ZnO HMS photoanode. For instance, the *PCE* of TiO_2_@ZnO HMS (5-min) reaches to 1.98% with *V*_oc_ = 0.41 V, *J*_sc_ = 10.49 mA cm^−2^, and FF = 0.46. Moreover, it is noteworthy that 10-min TiO_2_@ZnO HMS solar cell shows better performance with a *V*_oc_ of 0.46 V, a *J*_sc_ of 14.64 mA cm^−2^, and an *FF* of 0.47, yielding a *PCE* of 3.16%, which is much higher than the *PCE* of 1.54% obtained with blank photoanode. Figure 6 summarizes the relative variation of *PCE* and thickness of TiO_2_ passivation layer with different immersing time of ZnO HMS photoanode in TiCl_4_ aqueous solution. Obviously, the photovoltaic performance show an increase with rising immersing time, and *V*_oc_, *J*_sc_, FF, and PCE reach their maxima at 10 min. However, with further prolonging of immersing time, *J*_sc_ and *PCE* dramatically decrease at 15 min, with a decrease of *J*_sc_ from 14.64 mA cm^−2^ to 12.89 mA cm^−2^, and a decrease of *PCE* from 3.16% to 2.49%. In our series solar cells, the only difference is thickness of TiO_2_ passivation layer; therefore, it is reasonable to believe that effects related to light harvesting and electron transport resulted from adjusting of TiO_2_ passivation layer thickness are responsible for the variation of photovoltaic performance. 

To investigate the influence of TiO_2_ passivation layer on the optical absorption performance of ZnO HMS photoanode, Figure 7 displays the UV-vis absorption spectra of TiO_2_@ZnO HMS photoanodes obtained by immersing in TiCl_4_ aqueous solution for different periods. It is well-known that the absorbance and absorption range is related to the loading amount and band gap of QDs, respectively. As shown in Figure 7, the onset of absorption for all sample photoanodes is around 650 nm, indicating the typical band gap characteristic of CdSe QDs. However, the absorbance of the CdSe/CdS co-sensitized TiO_2_@ZnO HMS photoanode is higher than that of bare ZnO HMS photoanode, which means that a larger amount of CdSe/CdS QDs can be deposited on TiO_2_@ZnO HMS. According to Equation (1) [34], the light harvesting efficiency (*LHE*) is related to the absorbance.
*LHE*=1–10^-absorbance^,(1)

Apparently, with prolong treatment time in TiCl_4_ aqueous solution, the absorption intensity increases and reaches maximum value at 10 min. The increased absorbance from 0 to 10 min of immersing time can be ascribed to the improved surface stability and specific surface area, which can facilitate adsorption of more QDs [30,35]. Therefore, the *LHE* of 10-min TiO_2_@ZnO HMS photoanode is higher than bare ZnO HMS and other TiO_2_@ZnO HMS photoanodes. When immersing time prolonged to 15 min, a decrease of absorbance occurred because longer immersing time likely reduces the average pore size in the TiO_2_@ZnO HMS photoanode, which is unfavorable for QDs loading. As Equation (1) indicated, higher *LHE* obtained with 10-min TiO_2_@ZnO HMS photoanode means more photons are captured by the CdSe/CdS co-sensitized TiO_2_@ZnO HMS photoanode, contributing to the enhancement of *J*_sc_.

The significant enhancement of *J*_sc_ for our champion solar cell is a major key factor to improve photovoltaic performance. Under a given light source, *J*_sc_ is expressed by the Equation (2) [36]:(2)$$J_{sc}=q\int_{\lambda_{\min}}^{\lambda_{\max}}\eta_{IPCE}\phi_{ph}^{source}(\lambda )d\lambda ,$$
where *λ*_max_ and *λ*_min_ are the wavelengths where the *IPCE* vanishes, *Φ*_ph_ is the incident photon flux. Figure 8 compares the *IPCE* of QDSSCs based on TiO_2_-passivated ZnO HMS photoanodes with different treatment duration in TiCl_4_ aqueous solution. Two phenomena are displayed in IPCE results, one is that the similar photoresponse range to the profile of UV-vis spectra; another is that the *IPCE* values follow the order of 10 min > 15 min > 5 min > blank, which is in accordance with the corresponding changing trend of *J*_sc_. It is well-known that IPCE is determined by *LHE*, charge injection efficiency (*Φ*_ing_), and charge collection efficiency (*η*_cc_), as expressed in Equation (3) [37,38]:(3)$$IPCE=LHE\times \phi_{ing}\times \eta_{cc},$$

As discussed in Figure 7, higher *LHE* obtained with TiO_2_@ZnO HMS photoanode is one of the reasons responsible for the enhancement of *IPCE*. Moreover, the influence of TiO_2_ passivation layer on charge transfer may be another factor contributing to the enhancement of *IPCE*. Therefore, EIS was carried out to explore the kinetics of electrochemical and photoelectrochemical processes in the QDSSCs. Figure 9 gives the Nyquist plots of QDSSCs based on bare ZnO HMS and 10-min TiO_2_@ZnO HMS photoanodes, and the inset presents the equivalent circuit model for QDSSC. Two semicircles appear in EIS spectra. The first very small semicircle is assigned to the redox reaction at counter electrode/electrolyte interface at high frequencies (*R*_c_); the second larger semicircle reveals the charge transfer across the photoanode/QDs/electrolyte interface at intermediate frequency, and the size of the semicircle represents the charge transport/recombination resistance (*R*_ct_) [29,39]. Table 2 also summarizes the fitted results from EIS data by using the “Zview” software. (Zview 2, Solartron, London, UK).

The *R*_c_ of QDSSCs based on bare ZnO HMS is 5.1 Ω, which is similar to 4.8 Ω for cell based on TiO_2_@ZnO HMS because our QDSSCs employed the same counterelectrode and electrolyte. However, remarkable variation of *R*_ct_ is observed between QDSSC based on bare ZnO HMS (218 Ω) and 10-min TiO_2_@ZnO HMS (309 Ω). The increased recombination resistance as a result of the TiO_2_ passivation layer leads to lower electron recombination, which is due to the energy matching between ZnO and TiO_2_ which could hinder the back reaction of photo-generated electrons with oxidized species in polysulfide electrolyte. Moreover, lower recombination resistance caused by TiO_2_ passivation layer means a longer electron lifetime (*τ*_n_) can be expected. The *τ*_n_ is calculated by Equation (4) [40]:(4)$$\tau_{n}=C_{\mu }\times R_{ct},$$

As shown in Table 2, *τ*_n_ increases from 207 ms to 292 ms for the bare ZnO HMS and 10-min TiO_2_@ZnO HMS cells. A longer *τ*_n_ indicates a decreased charge recombination, proving that TiO_2_ passivation layer on ZnO HMS can greatly suppress the charge recombination in QDSSC. In addition, a reduced recombination will lead to the enhancement of *η*_cc_, not only resulting in the increase of *IPCE*, but also causing the increase of *V*_oc_ which is determined by the following Equation (5) [41]:(5)$$V_{oc}=\frac{E_{F}{}_{n}-E_{redox}}{e}=\frac{k_{B}T}{e}\ln(\frac{n}{n_{0}}),$$
where *E*_fn_ is the quasi-Fermi level of the electrons in semiconductor under illumination; *E*_redox_ is the potential of the redox electrolyte; *e* is the positive elementary charge; *k*_B_*T* is the thermal energy; *n* is the electron concentration in conduction band of the semiconductor under illumination; *n*_0_ is the electron concentration in the dark condition. As EIS results show, the TiO_2_ passivation layer caused a longer electron lifetime and improved *η*_cc_, which will lead to more electron accumulation in ZnO HMS photoanode, causing the *n* to increase, while the *E*_redox_ remains the same because of the identical polysulfide. Therefore, higher *V*_oc_ is obtained using TiO_2_-passivated ZnO HMS photoanode.

From UV-vis, IPCE, and EIS results, it is obvious that enhancement of *LHE* and decrease of charge recombination play decisive role in TiO_2_@ZnO HMS based solar cell for the improvement of *J*_sc_ and *V*_oc_. As discussed above, a possible mechanism for the contribution of TiO_2_ passivation layer toward improve QDSSC performance is proposed in Figure 10. First, the formation of TiO_2_ passivation layer can enhance the adsorption of QDs, improving the light capture ability. Second, the energy level between ZnO and TiO_2_ can prevent the back transfer of electrons in ZnO with polysulfide electrolyte, increasing electron lifetime. Therefore, it can be concluded that the increased light harvesting and charge collecting efficiency caused by TiO_2_ passivation layer lead to the improvement of *J*_sc_ and *V*_oc_, resulting in the superior photovoltaic performance of TiO_2_@ZnO HMS-based QDSSC. However, the lower *V***_oc_** and *FF* seem key factors that limit the further improvement of PCE, leading to our champion solar cell is still lower than the best solar cell reported in literature [1,4]. Some factors including the types of QDs, electrolyte, and counter electrode may have a role in enhancing the *V*_oc_ and *FF*. In future, we will pay more attention in developing new types of sensitizers or counter electrodes to realize the full utilization of light and band alignment to boost the *V*_oc_ and *FF* that may further help to enhance PCE of QDSSC based on TiO_2_-passivated ZnO HMS architectures.

## 4. Conclusions

In conclusion, we have synthesized ZnO HMS by a simple carbonaceous sphere template method, and further developed a feasible approach to passivated ZnO HMS with TiO_2_ layer for QDSSC application. TEM, SEM confirmed the structure of TiO_2_-passivated ZnO HMS and the thickness controllability of TiO_2_ layer by changing treatment time in TiCl_4_ aqueous solution. When used as photoanode in QDSSCs, TiO_2_-passivated ZnO HMS shows a positive influence on photovoltaic performance. UV-vis, IPCE, and EIS analysis results proved that TiO_2_ passivation layer could improve light harvesting efficiency, reduce charge recombination, enhance charge collecting efficiency, leading to the enhancement of *J*_sc_ and *V*_oc_. As a result, the *PCE* of QDSSC based on TiO_2_-passivated ZnO HMS with optimal thickness layer increased to 3.16%, which is much higher than 1.54% produced by QDSSC based on bare ZnO HMS.

## Figures and Tables

**Figure 1 nanomaterials-10-00631-f001:**
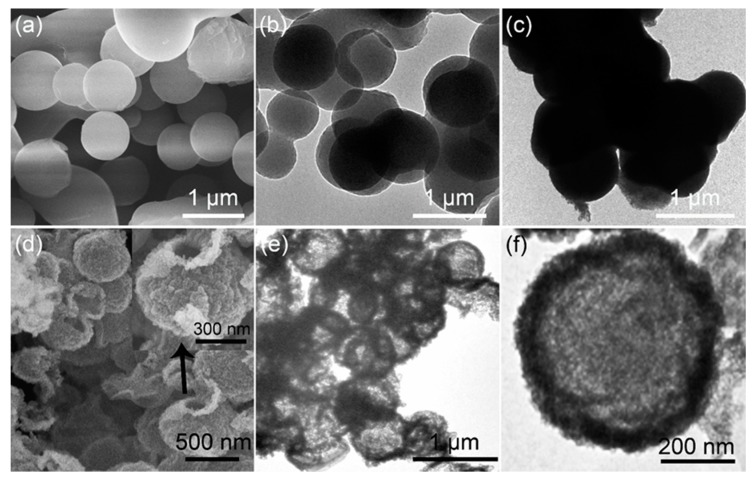
(**a**) SEM and (**b**) TEM of carbonaceous spheres templates; (**c**) TEM of carbonaceous spheres adsorbed Zn^2+^ precursors; (**d**) SEM of ZnO hollow microspheres (HMS); (**e**) low and (**f**) high magnification TEM of ZnO HMS.

**Figure 2 nanomaterials-10-00631-f002:**
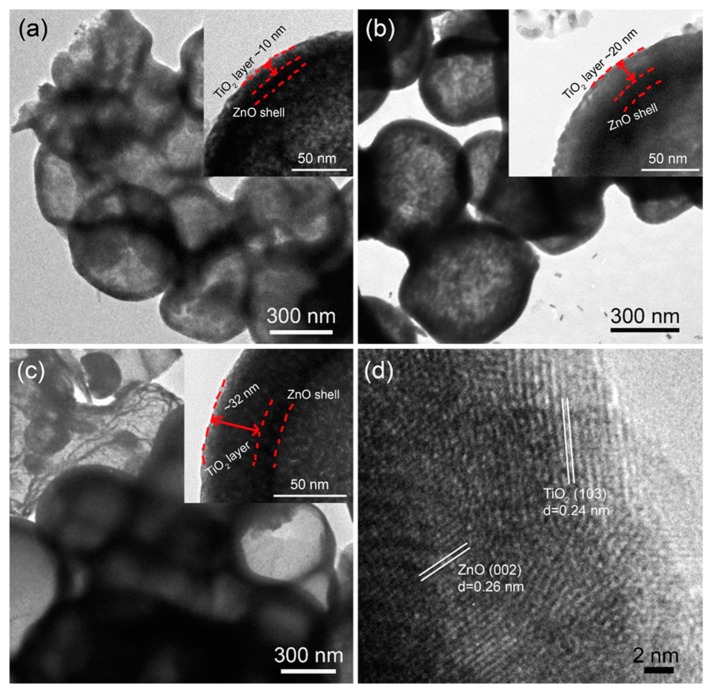
TEM images of TiO_2_-passivated ZnO HMS for different period (**a**) 5-min, (**b**) 10-min, and (**c**) 15-min; (**d**) high resolution TEM image of the TiO_2_-passivated ZnO HMS (10-min).

**Figure 3 nanomaterials-10-00631-f003:**
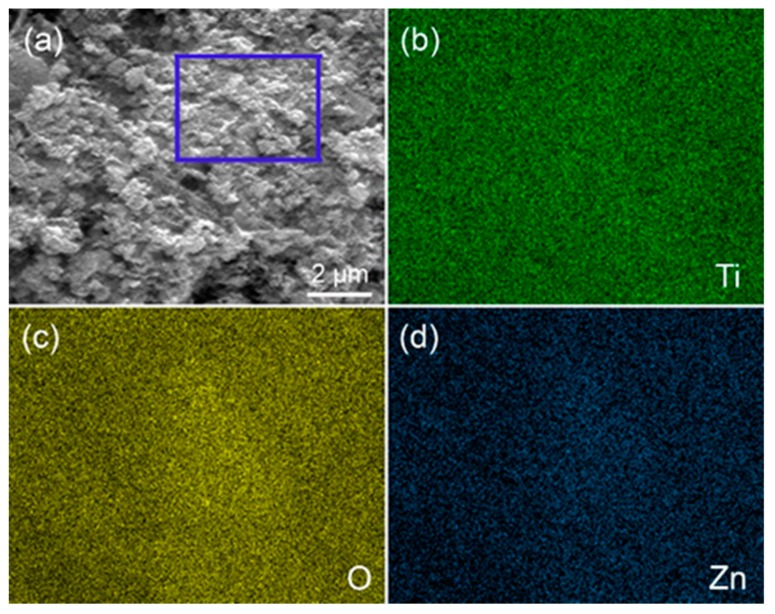
(**a**) The selected zone of for elemental mapping of TiO_2_-passivated ZnO HMS, (**b**–**d**) elemental mapping results of Ti, O, and Zn, respectively.

**Figure 4 nanomaterials-10-00631-f004:**
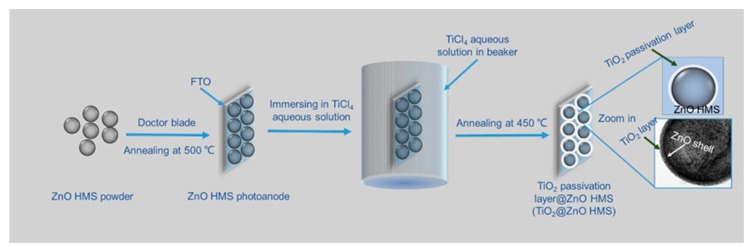
Illustration of TiO_2_-passivated ZnO HMS formation process.

**Figure 5 nanomaterials-10-00631-f005:**
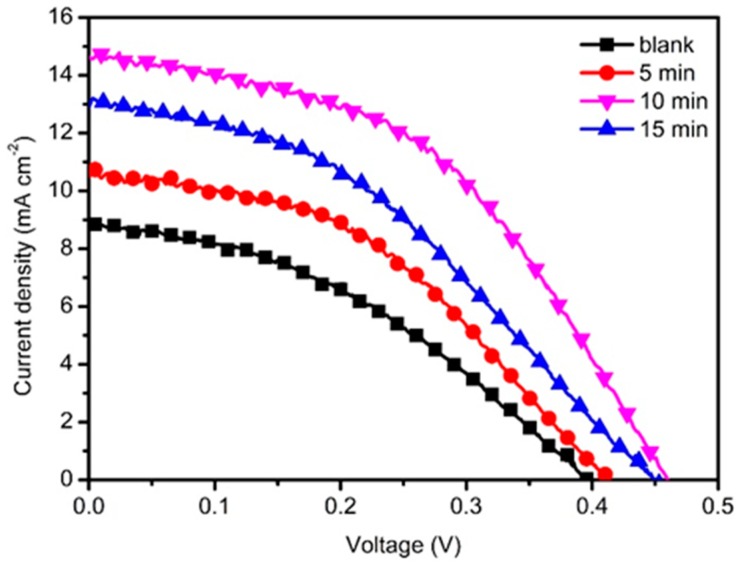
The *J*-*V* curves of QDSSCs based on TiO_2_-passivated ZnO HMS by immersing in TiCl_4_ aqueous solution for 0 min (blank), 5 min (5-min), 10 min (10-min), and 15 min (15-min), respectively.

**Figure 6 nanomaterials-10-00631-f006:**
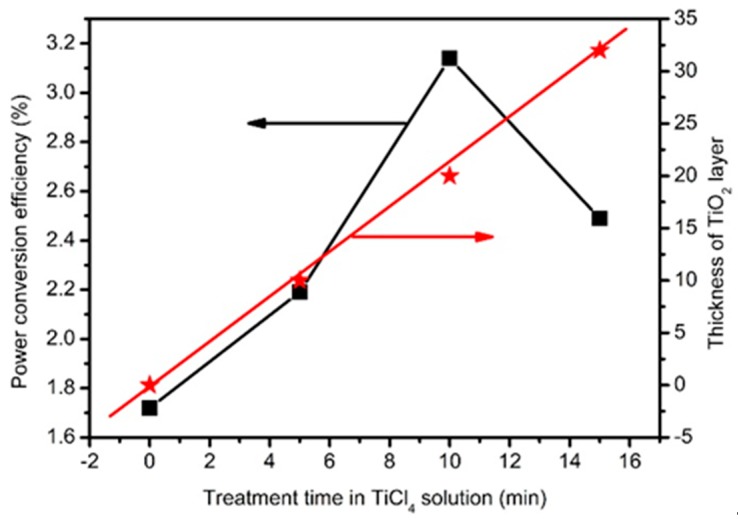
Relative variation of power conversion efficiency, and TiO_2_ passivation layer thickness with different immersing time.

**Figure 7 nanomaterials-10-00631-f007:**
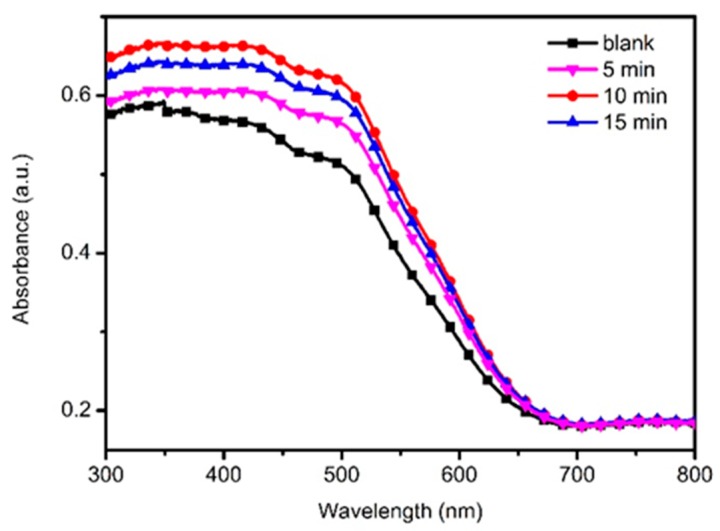
The UV-vis absorption spectra of photoanode based on TiO_2_-passivated ZnO HMS by immersing in in TiCl_4_ aqueous solution for 0 min (blank), 5 min (5-min), 10 min (10-min) and 15 min (15-min), respectively.

**Figure 8 nanomaterials-10-00631-f008:**
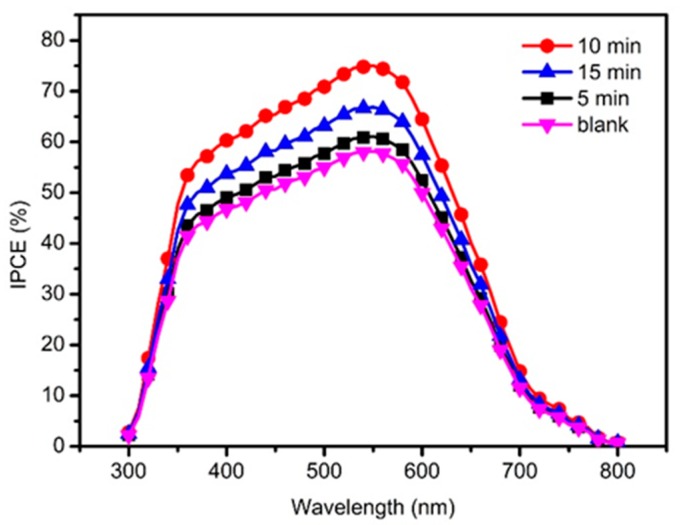
The *IPCE* spectra of photoanode based on TiO_2_-passivated ZnO HMS by immersing in TiCl_4_ aqueous solution for 0 min (blank), 5 min (5-min), 10 min (10-min) and 15 min (15-min), respectively.

**Figure 9 nanomaterials-10-00631-f009:**
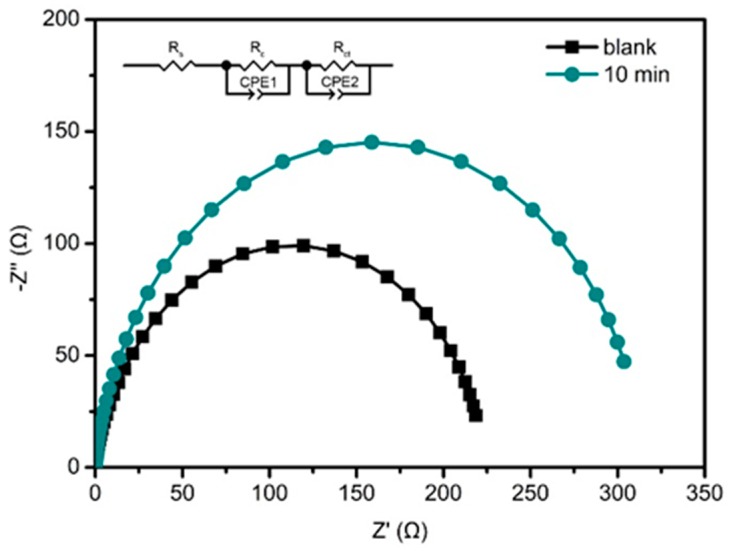
Nyquist plots of EIS spectra of QDSSCs recorded under dark at an applied bias of −0.5 V, the inset shows the equivalent circuit of our device.

**Figure 10 nanomaterials-10-00631-f010:**
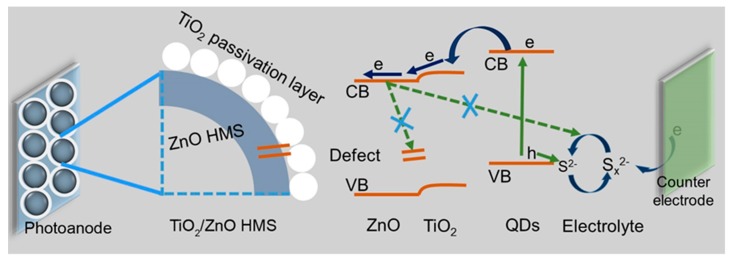
The illustration of charges deliver and recombination process on interfaces of TiO_2_@ZnO HMS-based QDSSC.

**Table 1 nanomaterials-10-00631-t001:** Average photovoltaic performance and standard deviation extracted from *J*-*V* curves of QDSSCs based on ZnO HMS passivated with TiO_2_ by immersing in TiCl_4_ aqueous solution for 0 min (blank), 5 min (5-min), 10 min (10-min), and 15 min (15-min) ^a^.

Solar cells	*V*_oc_ (V)	*J*_sc_ (mA cm^−2^)	*FF*	*PCE* (%)
Blank	0.39 (0.40)	8.75 (8.79)	0.43 (0.44)	1.49 ± 0.05 (1.55)
5-min	0.40 (0.41)	10.44 (10.49)	0.45 (0.46)	1.88 ± 0.12 (1.98)
10-min	0.45 (0.46)	14.57 (14.64)	0.46 (0.47)	2.99 ± 0.48 (3.16)
15-min	0.44 (0.45)	12.83 (12.89)	0.42 (0.43)	2.41 ± 0.10 (2.49)

^a^ The numbers in parentheses represent the values obtained from the champion solar cells.

**Table 2 nanomaterials-10-00631-t002:** Fitting results of EIS of QDSSCs based on ZnO HMS and TiO_2_@ZnO HMS photoanodes.

Photoanodes	*R*_c_ (Ω)	*R*_ct_ (Ω)	*C*_μ_ (μF)	*τ*_n_ (ms)
ZnO HMS	5.1	218	948	207
TiO_2_@ZnO HMS	4.8	309	945	292

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
