# Peer review of "TiO2 Passivation Layer on ZnO Hollow Microspheres for Quantum Dots Sensitized Solar Cells with Improved Light Harvesting and Electron Collection"

_nanomaterials, 2020, doi:10.3390/nano10040631_

Round 1
Reviewer 1 Report
The manuscript is devoted to electrode material suitable for quantum dots sensitized solar cells. The initial layer structure in the form of ZnO hollow microspheres was improved by formation core-shell structure consisting of ZnO microspheres coated by TiO2 layer. The efficiency of solar cells with such electrode has larger efficiency than for bare ZnO microspheres which are explained by better light absorption, the lower recombination rate of photogenerated carriers due to passivation of states on the electrode/active layer interface and longer electron lifetime. The subject is interesting from a scientific and applied point of view.
The manuscript is well organized, experimental methods are well-chosen and described.
I have just a few remarks.
_______________________
The expression “photoanode” is used in different ways trough the manuscript.
Abstract
“…ZnO HMS coated by TiO2 layer that is obtained by immersing ZnO HMS
photoanode in TiCl4 aqueous solution…”
Introduction
“…The photoanode is usually consisted of fluorine doped tin oxide (FTO) glass,
mesoporous metal oxide semiconductor (MOS) film, and quantum dots (QDs) [9, 10].”
________________________________________________
“the mixture was aged for 12 at room temperature.”
12 minutes?
-----------------------------------------------------------------------------------
”The ZnO hollow microspheres (HMS) are easily observed by TEM images as shown in Fig. 1(a) and (c).”
It is not truth.
_________________________________________________
183 “The surficial elemental composition”
“Surficial”?
226 “ It is well known that the absorbance is related to the loading amount and band gap of QDs.”
Unclear – what the authors wanted to say?
Author Response
Reviewer #1
The manuscript is devoted to electrode material suitable for quantum dots sensitized solar cells. The initial layer structure in the form of ZnO hollow microspheres was improved by formation core-shell structure consisting of ZnO microspheres coated by TiO2 layer. The efficiency of solar cells with such electrode has larger efficiency than for bare ZnO microspheres which are explained by better light absorption, the lower recombination rate of photogenerated carriers due to passivation of states on the electrode/active layer interface and longer electron lifetime. The subject is interesting from a scientific and applied point of view.
The manuscript is well organized, experimental methods are well-chosen and described.
I have just a few remarks.
Comment 1. The expression “photoanode” is used in different ways through the manuscript.
Abstract
“…ZnO HMS coated by TiO2 layer that is obtained by immersing ZnO HMS
photoanode in TiCl4 aqueous solution…”
Introduction
“…The photoanode is usually consisted of fluorine doped tin oxide (FTO) glass,
mesoporous metal oxide semiconductor (MOS) film, and quantum dots (QDs) [9, 10].”
Respond: we accept your comment. In Abstract, the “ZnO HMS photoanode” has been corrected to “ZnO HMS architectures”.
Comment 2. “the mixture was aged for 12 at room temperature.”
12 minutes?
Respond: We apologize for our carelessness and accept your comment. The typo has been corrected to “12 h” in the revised version.
Comment 3. “the ZnO hollow microspheres (HMS) are easily observed by TEM images as shown in Fig. 1(a) and (c).”
It is not truth.
Respond: We apologize for our confusing expression. In this revised manuscript, the sentence has be rewritten as “The ZnO hollow microspheres (HMS) could be indentified by TEM images as shown in Fig. 1(a) and (c).”
Comment 4. 183 “The surficial elemental composition”
“Surficial”?
Respond: We apologize for our carelessness, and the typo has been corrected to “surface elemental”.
Comment 5. 226 “ It is well known that the absorbance is related to the loading amount and band gap of QDs.”
Unclear – what the authors wanted to say?
Respond: We apologize for our confusing expression. In this revised version, the sentence has been rewritten, as “It is well known that the absorbance and absorption range is related to the loading amount and band gap of QDs, respectively.”

Reviewer 2 Report
The authors present a study of using TiO2 passivated ZnO hollow microspheres for quantum dot sensitized solar cells. In principle, solar cell research is of current large interest. However, the present study appears as a rather narrow technical investigation of the effect of the TiO2 passivation for a particular materials system.
For example, the authors present an impressive increase in efficiency by 100% relative. But the absolute value increases to just 3.16%. For the broader audience of Nanomaterials, it would be important to explain how such values stand in relation to other studies and fabricated solar cells.
When comparing to the currently fabricated commercial solar cell modules, where efficiencies of around 20% can be found, the presented cell efficiencyof around 3% for the present quantum-dot based solar cell appears very low and of limited interest. (Note that the highest reported single-junction solar cell efficiencies are at the level of 30%.)
It would be important to explain possible further steps to increase the efficiency, or why an increase is not needed. Currently, the study is of limited interest for the broad audience, unless a pathway to >10% efficiency can be shown.
That is, a more indepth investigation of what limits jsc, Voc, and FF should be presented. And possbile ways for increasing each of them should be demonstrated to motivate that the materials platform is viable for solar cells of applied interest.
Author Response
Reviewer #2
Comments and Suggestions for Authors
Comment 1. The authors present a study of using TiO2 passivated ZnO hollow microspheres for quantum dot sensitized solar cells. In principle, solar cell research is of current large interest. However, the present study appears as a rather narrow technical investigation of the effect of the TiO2 passivation for a particular materials system.
Respond: Thanks for your comment. ZnO is a potential candidate material for the photoanode due to its high electron mobility (200-1000 cm-2 V-1 S-1), however, the defects of ZnO and weak light scattering of common ZnO nanostructure such as nanoparticles, nanorods, and nanotube limits its application in QDSSC. Therefore, this study proposed a TiO2 layer passivated ZnO hollow microspheres architecture to enhance photovoltaic performance of QDSSC. By employing common CdS/CdSe quantum dots as sensitizer, the influence of proposed TiO2 passivated ZnO hollow microspheres structure on photovoltaic performance has been investigated by I-V, UV-vis, IPCE and EIS, showing that TiO2 passivated ZnO HMS architecture can yield good light harvesting, reduced charge recombination, and longer electron lifetime.
Comment 2. For example, the authors present an impressive increase in efficiency by 100% relative. But the absolute value increases to just 3.16%. For the broader audience of Nanomaterials, it would be important to explain how such values stand in relation to other studies and fabricated solar cells.
Respond: Thanks for your comment. In this work, the focus of our investigation is the effect of TiO2 passivated ZnO HMS on QDSSC. By using CdS/CdSe QDs, a champion PCE of 3.16% has been achieved based on this architecture, which is higher than previous report in reference [30]. This related description has been presented in Introduction as follows:
For example, Haifeng Zhao et al. [29] modified ZnO nanorods with TiO2 nanoparticles, and revealed that TiO2 passivation layer can facilitate the deposition of CdS/CdSe QDs on TiO2/ZnO surfaces, and effectively suppress the charge recombination. Lou et al. [30] reported a PCE of 1.97% for QDSSCs based on ZnO nanorods passivated with TiO2 as a barrier layer. In view of these backgrounds, it can be found that ZnO surface modification with TiO2 has attracted interest of researchers to enhance photovoltaic performance of QDSSCs, due to the combination of excellent electron mobility of ZnO and high chemical stability of TiO2 [29].
References
- Zhao, H.; Wu, Q.; Hou, J.; Cao, H.; Jing, Q.; Wu, R.; Liu, Z. Enhanced light harvesting and electron collection in quantum dot sensitized solar cells by TiO2 passivation on ZnO nanorod arrays. Sci. China Mater. 2017, 60, 239-250.
- Lou, Y.; Yuan, S.; Zhao, Y.; Hu, P.; Wang, Z.; Zhang, M.; Shi, L.; Li, D. A simple route for decorating TiO2 nanoparticle over ZnO aggregates dye-sensitized solar cell. Chem. Eng. J. 2013, 229, 190-196.
Comment 3. When comparing to the currently fabricated commercial solar cell modules, where efficiencies of around 20% can be found, the presented cell efficiency of around 3% for the present quantum-dot based solar cell appears very low and of limited interest. (Note that the highest reported single-junction solar cell efficiencies are at the level of 30%.)
Respond: Thanks for your comment. It is truce that our solar cell’s efficiency is much lower than single-junction solar cell. However, QDSSC is not the same type solar cell comparing with single-junction solar cell. In fact, the efficiency of QDSSC varied much if different types of QDs are used. To our best knowledge, in just CdS/CdSe QDs system, the efficiency over 3% can be classified to high category, because very recent report efficiency of CdSe/CdSe QDSSC reached around 4% (See reference [29]). Furthermore, the focus of our investigation is the effect TiO2 passivated ZnO HMS architecture in QDSSC, the CdS/CdSe quantum dots is just a sensitizer we selected to reveal that our architecture has the function to improve photovoltaic performance. We believe that our architecture is suitable for other types of sensitizers such as dyes or perovskites, and may achieve better photovoltaic performance employing these sensitizers.
Reference
- Zhao, H.; Wu, Q.; Hou, J.; Cao, H.; Jing, Q.; Wu, R.; Liu, Z. Enhanced light harvesting and electron collection in quantum dot sensitized solar cells by TiO2 passivation on ZnO nanorod arrays. Sci. China Mater. 2017, 60, 239-250.
Comment 4. It would be important to explain possible further steps to increase the efficiency, or why an increase is not needed. Currently, the study is of limited interest for the broad audience, unless a pathway to >10% efficiency can be shown.
Respond: Thanks for your comment. At present, develop a QDSSC with efficiency over 10% is still a challenge for our group, because our group are most interest in metal oxide structure design rather than synthesis of new type of QDs as sensitizer, that is why we just borrow classical CdS/CdSe quantum dots to sensitized TiO2 passivated ZnO HMS for solar cell application. However, we present an impressive increase in efficiency by 100% using proposed TiO2 passivated ZnO HMS, demonstrating that this architecture is an effective strategy to boost photovoltaic performance in sensitized solar cell. We believe that our architecture may achieve better photovoltaic performance if some new types of sensitizer is employed such as perovskites.
Comment 5. That is, a more in depth investigation of what limits jsc, Voc, and FF should be presented. And possbile ways for increasing each of them should be demonstrated to motivate that the materials platform is viable for solar cells of applied interest.
Respond: Thanks for your comment. In this revised version, we analyze the possible reasons limit the Voc and FF as suggested, which is presented in revised version as follows:
However, the lower Voc and FF seem key factors that limit the further improvement of PCE, leading to our champion solar cell is still lower than the best solar cell reported in literature [1, 4]. Some factors including the types of QDs, electrolyte, and counter electrode may have influence on enhancing the Voc and FF. In future, we will pay more attention in developing new types of sensitizers or counter electrodes to realize the full utilization of light and band alignment to boost the Voc and FF, that may further help to enhance PCE of QDSSC based on TiO2 passivated ZnO HMS architectures.
References
- Chebrolu,V.T.; Kim,H.-J. Recent progress in quantum dot sensitized solar cells: an inclusive review of photoanode, sensitizer, electrolyte, and the counter electrode. J. Mate. Chem. C 2019, 7, 4911-4933.
- Yang, X.; Wang, H.; Cai, B.; Yu, Z.; Sun, L. Progress in hole-transporting materials for perovskite solar cells. J. Energy Chem. 2018, 27, 650-672.

Reviewer 3 Report
Paper is interesting and should be published after revision.
- Please add error for PV parameters in Table 1 along with in formation how much devices were constructed and tested. Please add photo of created solar cells.
- Please add in Introduction information about various method of obtained ZnO such as Atomic Layer Depositions (ALD) and mention about other papers with ZnO as interlayer in solar cells, e.g. Electrochimica Acta 191 (2016) 784–794.
- Please explain why do you used as precursor TiCl4 to obtained TiO2 layer. What about other precursors, e.g. TIPO.
- Why do you proposed immersion to create TiO2 layer? Maybe it will be more useful used spin-coating or doctor-blade technique. Please mention about it in paper.
- Calcining process at 450 C suggested that you should obtained TiO2 in anatase form. Please confirmed crystallographic structure of TiO2 by XRD.
Author Response
Reviewer #3
Paper is interesting and should be published after revision.
Comment 1. Please add error for PV parameters in Table 1 along with information how much devices were constructed and tested. Please add photo of created solar cells.
Response: Thanks for your comment and we fully accept your suggestion. Actually, each of QDSSC has been repeatedly tested three times in our investigation. In this revised manuscript, three times tested J-V curves CdSe/CdS QDSSCs based on ZnO HMS passivated with TiO2 by immersing in TiCl4 aqueous solution for 0 min (blank), 5 min (5-min), 10 min (10-min) and 15 min (15-min) are provided Figure S2 in Supplementary Materials. A photo of created solar cell has been also added as an inset of Figure S2. The photovoltaic performance of solar cells, including short-circuit current density (Jsc), open voltage (Voc), fill factor (FF), power conversion efficiency (PCE), and their corresponding average value and standard deviation are also present in Table S1-Table S4 in Supplementary Mateials. Furthermore, the Table 1 summarizes the average photovoltaic performance and standard deviation extracted from J-V curves.
Figure S2. (a-d) J-V curves CdSe/CdS QDSSCs based on ZnO HMS passivated with TiO2 by immersing in TiCl4 aqueous solution for 0 min (blank), 5 min (5-min), 10 min (10-min) and 15 min (15-min), each solar cell have been repeatedly tested three times; the inset is a photograph of assembled solar cell in an open way for J-V test.
Table S1. Average photovoltaic performance and standard deviation extracted from J-V curves of QDSSCs based on ZnO HMS passivated with TiO2 by immersing in TiCl4 aqueous solution for 0 min (blank).
|
Voc (V) |
Average Voc (V) |
Jsc (mA cm-2) |
Average Joc (mA cm-2) |
FF |
Average FF |
PCE (%) |
Average PCE (%) |
Standard Deviation 0f PCE |
|
0.40 |
0.393 |
8.79 |
8.750 |
0.44 |
0.433 |
1.547 |
1.492 |
0.048 |
|
0.39 |
8.72 |
0.43 |
1.462 |
|||||
|
0.39 |
8.74 |
0.43 |
1.466 |
Table S2. Average photovoltaic performance and standard deviation extracted from J-V curves of QDSSCs based on ZnO HMS passivated with TiO2 by immersing in TiCl4 aqueous solution for 5 min.
|
Voc (V) |
Average Voc (V) |
Jsc (mA cm-2) |
Average Joc (mA cm-2) |
FF |
Average FF |
PCE (%) |
Average PCE (%) |
Standard Deviation 0f PCE |
|
0.41 |
0.40 |
10.49 |
10.440 |
0.46 |
0.450 |
1.978 |
1.880 |
0.123 |
|
0.40 |
10.43 |
0.46 |
1.919 |
|||||
|
0.39 |
10.39 |
0.43 |
1.742 |
Table S3. Average photovoltaic performance and standard deviation extracted from J-V curves of QDSSCs based on ZnO HMS passivated with TiO2 by immersing in TiCl4 aqueous solution for 10 min.
|
Voc (V) |
Average Voc (V) |
Jsc (mA cm-2) |
Average Joc (mA cm-2) |
FF |
Average FF |
PCE (%) |
Average PCE (%) |
Standard Deviation 0f PCE |
|
0.46 |
0.447 |
14.64 |
14.570 |
0.47 |
0.460 |
3.165 |
2.995 |
0.483 |
|
0.44 |
14.58 |
0.46 |
2.951 |
|||||
|
0.44 |
14.49 |
0.45 |
2.869 |
Table S4. Average photovoltaic performance and standard deviation extracted from J-V curves of QDSSCs based on ZnO HMS passivated with TiO2 by immersing in TiCl4 aqueous solution for 15 min.
|
Voc (V) |
Average Voc (V) |
Jsc (mA cm-2) |
Average Joc (mA cm-2) |
FF |
Average FF |
PCE (%) |
Average PCE (%) |
Standard Deviation 0f PCE |
|
0.45 |
0.443 |
12.89 |
12.827 |
0.43 |
0.423 |
2.494 |
2.406 |
0.098 |
|
0.45 |
12.82 |
0.42 |
2.423 |
|||||
|
0.43 |
12.77 |
0.42 |
2.301 |
Table 1. Average photovoltaic performance and standard deviation extracted from J-V curves of QDSSCs based on ZnO HMS passivated with TiO2 by immersing in TiCl4 aqueous solution for 0 min (blank), 5 min (5-min), 10 min (10-min) and 15 min (15-min) a.
|
Solar cells |
Voc (V) |
Jsc (mA cm-2) |
FF |
PCE (%) |
|
Blank |
0.39 (0.40) |
8.75 (8.79) |
0.43 (0.44) |
1.49±0.05 (1.55) |
|
5-min |
0.40 (0.41) |
10.44 (10.49) |
0.45 (0.46) |
1.88±0.12 (1.98) |
|
10-min |
0.45 (0.46) |
14.57 (14.64) |
0.46 (0.47) |
2.99±0.48 (3.16) |
|
15-min |
0.44 (0.45) |
12.83 (12.89) |
0.42 (0.43) |
2.41±0.10 (2.49) |
|
a The numbers in parentheses represent the values obtained from the champion solar cells. |
||||
Comment 2. Please add in Introduction information about various method of obtained ZnO such as Atomic Layer Depositions (ALD) and mention about other papers with ZnO as interlayer in solar cells, e.g. Electrochimica Acta 191 (2016) 784–794.
Respond: we accept your comment, and thank you for your recommendation of literatures on ZnO preparation method. In this revised manuscript, the information about methods of obtained ZnO has been introduced, and the recommended reference has been cited. The corresponding text are added as follows:
Among them, ZnO is a potential candidate material for the photoanode due to its high electron mobility (200-1000 cm-2 V-1 S-1) [15] and various fabrication strategies such as atomic layer deposition (ALD) and hydrothermal method, for variety of structures including 1D, 2D, and 3D [16-18].
References
- Agnieszka, I.; Marcin, P.; Igor, T.; Bartosz, B.; Rafal, P.; Michal, F.; Jacek, W.; Bartłomiej Sławomir, W.; Filip, G.; Marek, G. Influence of ZnO: Al, MoO3 and PEDOT: PSS on efficiency in standard and inverted polymer solar cells based on polyazomethine and poly(3-hexylthiophene). Electrochim. Acta 2016, 191, 784-794.
Comment 3. Please explain why do you used as precursor TiCl4 to obtained TiO2 layer. What about other precursors, e.g. TIPO.
Respond: Thanks for your comment. The reason we use TiCl4 aqueous solution as precursor is that Ti4+ will be easily adsorbed on the suface of ZnO HMS just by immersing ZnO HMS in TiCl4 aqueous solution without any other chemical assistance.
Comment 4. Why do you proposed immersion to create TiO2 layer? Maybe it will be more useful used spin-coating or doctor-blade technique. Please mention about it in paper.
Respond: thanks for your comment. In fact, the ZnO hollow microspheres (HMS) layer is fabricated on surface of FTO by doctor blading technique in our experiment. Our final aim is to synthesis TiO2 passivated ZnO HMS as architecture for QDSSC, which need the TiO2 layer encapsulate the ZnO HMS. However, spin-coating or doctor-blade technique is only suitable to create thin film, cannot guarantee the encapsulation of each ZnO HMS with TiO2. By immersion of ZnO HMS in TiCl4 aqueous solution will allow the adsorption of Ti4+ precursor around the surface of each ZnO HMS, when annealing in air, the Ti4+ precursor adsorbed on surface of ZnO HMS will be oxidized into TiO2, forming the TiO2 layer passivated ZnO HMS architecture. Therefore, the immersion method is a feasible way to produce TiO2 passivated ZnO HMS.
Comment 5. Calcining process at 450 °C suggested that you should obtained TiO2 in anatase form. Please confirmed crystallographic structure of TiO2 by XRD.
Respond: We fully accept your comment. In this revised version, the XRD of TiO2 passivated ZnO HMS has been provided in Figure S1 in Supplementary Materials. The corresponding discussion are listed as follows:
Figure S1 provides the XRD of TiO2 passivated ZnO HMS, apart from diffraction peaks that can be ascribed to the hexagonal ZnO, the diffraction peaks around 2θ=24.6° and 53.1° belongs to the anatase phase of TiO2 (See Supplementary Materials), confirming the anatase TiO2 layer formation on ZnO HMS.
Figure S1. The XRD pattern of ZnO HMS passivated with TiO2 layer

Round 2
Reviewer 2 Report
The authors make a compelling case for why their work of the TiO2 passivation layer should merit publication.